# WRKY Transcription Factor Response to High-Temperature Stress

**DOI:** 10.3390/plants10102211

**Published:** 2021-10-18

**Authors:** Zhuoya Cheng, Yuting Luan, Jiasong Meng, Jing Sun, Jun Tao, Daqiu Zhao

**Affiliations:** 1College of Horticulture and Plant Protection, Yangzhou University, Yangzhou 225009, China; zhuoya.edu@outlook.com (Z.C.); jsmeng@yzu.edu.cn (J.M.); jingsun@yzu.edu.cn (J.S.); taojun@yzu.edu.cn (J.T.); 2Joint International Research Laboratory of Agriculture and Agri-Product Safety, The Ministry of Education of China, Yangzhou University, Yangzhou 225009, China; luanyuting96@gmail.com

**Keywords:** high-temperature stress, WRKY TFs, gene response, signaling pathway

## Abstract

Plant growth and development are closely related to the environment, and high-temperature stress is an important environmental factor that affects these processes. WRKY transcription factors (TFs) play important roles in plant responses to high-temperature stress. WRKY TFs can bind to the W-box *cis*-acting elements of target gene promoters, thereby regulating the expression of multiple types of target genes and participating in multiple signaling pathways in plants. A number of studies have shown the important biological functions and working mechanisms of WRKY TFs in plant responses to high temperature. However, there are few reviews that summarize the research progress on this topic. To fully understand the role of WRKY TFs in the response to high temperature, this paper reviews the structure and regulatory mechanism of WRKY TFs, as well as the related signaling pathways that regulate plant growth under high-temperature stress, which have been described in recent years, and this paper provides references for the further exploration of the molecular mechanisms underlying plant tolerance to high temperature.

## 1. Introduction

With the ever-increasing global temperatures caused by climate change [1], high temperature has become an important factor that affects plant growth [2,3], including crop yield, fruit quality, flower opening and ornamental value [1]. It has been proven that under high-temperature stress, changes in plant morphology, physiology and biochemistry occur [4]. The morphological consequences of high-temperature stress include leaf senescence and shedding, bud and root growth inhibition, and fruit discoloration and damage, and plants may die in a short time [1]. In addition, if the temperature is too high, plant respiration increases, and the regulation of photosynthesis is disrupted. For example, when a plant is in a dry state or a particular development period, high-temperature environments shorten the entire growth cycle and affect light induction and carbon assimilation (photosynthesis, transpiration, etc.) processes, which leads to reduced crop yields [5]. Moreover, excessive levels of reactive oxygen species (ROS) accumulate, leading to oxidative stress and cell death [6,7,8]. Accordingly, understanding how plants respond to high-temperature stress and its consequences has received extensive attention from plant researchers [9].

To resist the damage caused by high-temperature environments, plants have developed a series of mechanisms to survive adversity. Many transcription factors (TFs), such as MYB, AP2/ERFBP, NAC and WRKY, are involved in the response to abiotic stresses [10]. Plant-specific WRKY TFs are a family of genes that play important roles in many different pathways that respond to diverse abiotic stresses (cold, drought, UV-B, high temperature, salt and alkaline environments) [11,12,13,14]. Research has shown that overexpression of *AtWRKY57* can increase the drought tolerance of *Arabidopsis* [15]. *GhWRKY17* expression in cotton can be induced by drought, salt and H_2_O_2_ [16]. UV-B radiation treatment induces the expression of three *AtWRKYs* in *Arabidopsis* and *OsWRKY89* in rice, resulting in the production of a thick waxy substance on the leaf surface and improved tolerance to high temperature [17,18]. The WRKY family is known to be involved in the high-temperature response, when plants are subjected to high-temperature stress, WRKY TFs regulate gene expression through multiple pathways and at various levels, thereby regulating plant stress response signaling [19,20]. In the present review, we have first summarized the structural characteristics and classification of WRKY TFs, and provided a more complete and systematic review of the role of WRKY TFs in high-temperature environments on plants, focusing on the aspects of regulatory mechanisms and signaling pathways.

## 2. Characteristics of WRKY TFs

### 2.1. Structure and Classification

WRKY TFs are DNA-binding proteins that have a highly conserved approximately 60-amino-acid domain. The N-terminus is a conserved 7-peptide WRKYGQK motif, which is the core region, and the C-terminus is a zinc-finger structure C_2_H_2_ (CX_4–5_CX_22–23_HXH) or C_2_HC (CX_7_CX_23_HXC) [13,21]. In a few WRKY proteins, the WRKY domain is replaced by WRRY, WSKY, WKRY, WVKY or WKKY motifs [22]. The zinc-finger structure is essential for DNA binding by WRKY proteins [23], and the W-box TTGAC (C/T) promoter of two cDNAs encoding proteins from *Saccharomyces cerevisiae* (ABF1 and ABF2) is replaced by an additional divalent metal ion chelating agent, namely, 1,10-phenanthroline; these results indicate that WRKY proteins have a zinc-finger structure [24]. Yamasaki et al. [25] first resolved structural problems associated with the WRKY domain. The WRKY domain consists of four β-sheets and a zinc-binding pocket formed by the binding of zinc ions to histidine or cysteine residues. The N-terminal residue of WRKYGQK protrudes from the surface of the protein, allowing the approximately 6-bp DNA groove formed by the WRKY domain to interact with DNA; this is induced by the C-terminus WRKY domain. Therefore, WRKY proteins can interact with clusters of W-boxes (with the core motif TTGACC/T) and promoters of downstream genes to regulate the dynamic network of signals and their related physiological responses [25,26,27].

According to the number of WRKY domains and the zinc-finger structure, WRKY proteins can be divided into group I (two DBDs and one C_2_H_2_ zinc-finger structure), group II (a single DBD with different C_2_H_2_ zinc fingers) and group III (a single DBD with C_2_HC zinc fingers) [10]. The zinc-finger structure of group I and group II is CX_4-5_CX_22-23_HxH, and the nonsingle group II is divided into IIa, IIb, IIc, IId and IIe according to the primary amino acid sequence [23,28,29] (Figure 1 and Figure 2). The zinc-finger structure of group III is CX_7_CX_23_HxC. However, there are also a few structural types of WRKY proteins that do not match these three characteristic types, such as AtWRKY10 in *Arabidopsis*, of which the structure may be characterized by the lack of a conserved N-terminal WRKY domain and the inclusion of only one conserved WRKY domain that is similar to but inconsistent with the structural characteristics of group I proteins [30]. Group III is expressed only in higher plants, and most of these proteins are related to plant responses to biological stress, whereas group I is expressed not only in higher plants but also in ferns and some eukaryotic cells, such as myxomycetes and unicellular protozoa [13,31]. This indicates that WRKY TFs may originate from eukaryotic cells, and group I is the original form of these proteins.

### 2.2. Origin and Evolution

Comparative genome analysis shows that a large number of transcription regulator genes are present in the genomes of animals and plants. The evolution and diversity of eukaryotes appear to be related to the expansion of lineage-specific families [32]. Genes encoding WRKY proteins have been found in some nonphotosynthetic eukaryotes [33]. WRKY proteins have recently been identified as transcriptional regulators that include a large gene family [34]. The first cDNA SPF1 encoding a WRKY protein was cloned from sweet potato [35]. Then, many WRKY proteins were identified in different species, including *Arabidopsis* [36] (Figure 1), rice [37,38], wheat [37], wild oats [24], potato [39,40] and orchard grass [41]. In lower plants, WRKY family genes are found in organisms such as *Giardia*
*lamblia*, *Dictyostelium discoideum* and *Chlamydomonas reinhardtii* [28], and *WRKY*-expressed sequence tags (ESTs) have been identified in ferns and mosses [13].

## 3. Regulatory Mechanism of WRKY TFs

### 3.1. Transcription-Level Regulation

#### 3.1.1. WRKY Expression Patterns

TF expression patterns are generally divided into constitutive and inducible expression, and *WRKYs* are TFs of which the expression is inducible in response to various stresses. At present, *WRKY* expression is known to be affected by various environments (such as pathogens, signal molecules, temperature, drought and mechanical stress), and its expression is rapid, immediate and tissue specific; additionally, WRKYs participate in various physiological processes in plants [42]. WRKY TFs constitute the second-largest group in the senescent leaf transcriptome of *Arabidopsis* [43,44]. The expression of several *WRKYs* in *Arabidopsis* is strongly upregulated during plant senescence [13]. In a recent study, various *cis*-acting elements related to stress and defense responses were found in the promoter region of camellia *CsWRKYs*, and the study showed that the expression of 56 *CsWRKYs* was induced by abiotic stress [45]. Moreover, the stress response and tissue specificity of *WRKYs* in two wild potato varieties were investigated for the first time, and six and 11 *WRKYs* were identified as flower- and leaf-specific genes, respectively [46,47]. In addition, potato genes *StWRKY016*, *StWRKY045* and *StWRKY055* appear to be necessary for plants to cope with high-temperature stress, and they are leaf-specific [48]. Further studies of *AtWRKY25* showed that, during high-temperature stress, the expression levels of *AtWRKY25* high-temperature-induced genes or two oxidative stress response genes were more or less down-regulated [49]. Studies also showed that five *TaWRKYs* responded to high-temperature stress at different growth stages in wheat, and their expression levels might be upregulated or downregulated [50]. These results indicate that *WRKY* expression is not only induced by various environmental factors but also exhibits tissue-specificity.

#### 3.1.2. WRKY TFs Interact with Downstream Target Genes

Gene expression regulated by WRKY proteins mainly occurs via the DNA-binding specific *cis*-regulatory element TTGACC/T (W-box) [34,51], and WRKY proteins can activate or inhibit downstream gene transcription [52]. Many gel electrophoresis, random binding site selection, yeast one-hybrid and cotransfection experiments on WRKY TFs have shown that the W-box is a short sequence located in the promoter region of some genes. Previous studies have shown that CaWRKY6 of the IIb subgroup in pepper could bind to and activate the promoter region of *CaWRKY40*; thus, *CaHIR1*, *CaDEF1*, *CaPO2*, *CaHSP24* and other defense-related genes are common targets of CaWRKY6 and CaWRKY40 [20]. Moreover, the rice *HSP101* promoter is a high temperature-inducible promoter, *OsWRKY11*, of which the overexpression is induced by high temperature, bound to the promoter region of *OsHSP101* to enhance the tolerance of transgenic rice seedlings to high temperature [14,53] (Table 1). Additionally, experiments have proven that WRKY TFs can self-regulate and mutually regulate their own expression by directly binding to their own promoter sequence. *Arabidopsis AtWRKY53* directly binds to the *AtWRKY42* promoter and has a negative regulatory effect on leaf senescence [54,55].

### 3.2. Protein-Level Regulation

There are few reports about WRKY-interacting proteins in plants under high-temperature stress. It has been established that proteins, especially regulatory proteins, rarely act alone. Usually, they interact with each other temporarily or permanently and perform biological functions in biological systems. The interaction between TFs and signal proteins is a common mechanism for signal transduction to the nucleus, and studies in the past few years have shown that WRKY TFs interact with various proteins involved in signal transcription and chromatin remodeling. In addition, the WRKY domain and other protein motifs participate in interactions between proteins and mediate complex functional interactions [77].

#### 3.2.1. WRKY TFs Interact with Other WRKY Proteins

Interactions can occur between WRKY proteins. In *Arabidopsis*, AtWRKY30 can interact with AtWRKY54 and AtWRKY70. It has ben speculated that AtWRKY30 may participate in the aging regulatory network [78]. Yang et al. [79] found that JrWRKY2 and JrWRKY7 in walnut could form homodimers or heterodimers in response to abiotic stress.

#### 3.2.2. WRKY-VQ Motif Protein Interaction

The VQ motif-containing protein is a type of plant-specific transcriptional regulatory factor that plays an important role in plant growth and development and in response to abiotic stress [80]. In the past few years, several teams have reported the interaction of AtWRKY proteins with a class of proteins containing the poly-amino acid sequence FxxxVQxLTG (VQ element). Since the β-sheet of the WRKY domain is in direct contact with DNA bases, the VQ protein can directly change the conformation of the WRKY DNA-binding domain to affect the DNA binding of the interacting WRKY protein [25,81]. It has been reported that the overexpression of some soybean *GmVQs* in *Arabidopsis* has a strong influence on high-temperature stress in plants [82]. A study showed that ectopic expression of tomato *SlVQ6* in *Arabidopsis* decreased tolerance to high temperature [83]. Studies have shown that AtWRKY8 can interact with AtVQ10 and AtVQ11 in response to high-temperature stress [57]. As discussed earlier, WRKYs tend to interact with different VQ proteins, which are hypothesized to induce histone modifications and chromatin remodeling to regulate the expression of downstream genes. Thirty-four AtVQ proteins that belong to subfamilies I and IIc in *Arabidopsis* can interact with AtWRKY proteins [84].

#### 3.2.3. WRKY TFs Interact with Other Proteins

In recent years, a large number of WRKY-interacting proteins have been identified. With the identification of histone deacetylase (HDAC) and histones as WRKY-interacting proteins, researchers have also begun to reveal the mechanism by which chromatin remodeling regulates target gene transcription [77]. In *Arabidopsis*, AtWRKY33 also interacts with proteins related to autophagy 18a (ATG18a), which is a key component of autophagy. In plants, autophagy participates in the nutrient cycle and responds to a series of abiotic stresses [85]. In addition, the WRKY protein MaWRKY1 from bananas interacts with the linker histone H1, and MaHIS1 may be related to fruit ripening and stress responses and may be functionally coordinated with MaWRKY1 in these physiological processes [86].

Studies have shown that WRKY TFs can combine with calmodulin (CaM) to regulate plant tolerance to high temperature. CaM is a Ca^2+^-binding signal protein that is expressed in all eukaryotes. The main function of CaM is to change the interaction of Ca^2+^ signals by binding ions and then change its interaction with various target proteins. An *Arabidopsis* cDNA expression library was screened with CaM as a probe, and a positive clone encoding the WRKY protein AtWRKY7 of group IId was isolated [87]. WRKY IId TFs in *Arabidopsis* (WRKY7/11/15/17/21/39/74) all contain a highly conserved CaM-binding site (CaMBD) that can bind to CaM in vitro [56]. CaM can bind to AtWRKY53 in *Arabidopsis* and affect downstream gene regulation [88].

## 4. WRKY TFs in Response to High Temperature

### 4.1. Role of WRKY TFs under High-Temperature Stress Conditions

An increasing number of studies have shown that WRKY TFs can positively respond to the tolerance of plants to high temperatures. For example, *CaWRKY40* in pepper participates in the response of plants to high-temperature stress [89], and the overexpression of *CaWRKY40* reduces the sensitivity of tobacco to high-temperature treatment, whereas the loss of this TF reduces this tolerance [90]. Similar results were found for *CaWRKY40* and *GhWRKY39*, expressed in upland cotton [91,92]. Moreover, the overexpression of wheat *TaWRKY1* and *TaWRKY33* in *Arabidopsis* activated downstream stress response-related genes to regulate tolerance to high temperature [52,53]. *AtWRKY25*, *AtWRKY26* and *AtWRKY33* play important roles in high temperature resistance (Figure 2) (Table 1) [61]. High temperature treatment inhibited the expression of *AtWRKY33* and induced the expression of *AtWRKY25* and *AtWRKY26* in *Arabidopsis*, and the constitutive overexpression of *AtWRKY25* and *AtWRKY26* enhanced resistance to high-temperature stress [58], and the silencing of *AtWRKY41* expression reduced seed dormancy and high temperature inhibition [61]. In wheat, the overexpression of *TaWRKY30* led to the enhancement of high-temperature tolerance [59]. However, there are also WRKYs in plants that play negative roles in response to high-temperature tolerance; for example, the heterologous expression of sunflower *HaWRKY6* in *Arabidopsis* could significantly reduce its tolerance to high temperature [70]. These studies indicate that WRKY TFs may improve the tolerance of plants to high temperature through transcriptional regulation.

### 4.2. Role of WRKY TFs in Signaling in Response to High Temperature

WRKY TFS participate in ROS signaling, plant hormone signal transduction pathways, and the mitogen-activated protein kinase (MAPK) cascade in response to abiotic stress. These reactions are intertwined with the interactions of many plant hormones, calcium and different ROS, as well as a large number of receptors; related enzymes; phosphatases; and other regulatory proteins, compounds and small molecules [93,94]. Plants can adjust their biological activities according to various environmental and internal signals. After a plant receptor protein recognizes a signal, the activity of the receptor protein will change, thus regulating the intracellular signal transduction pathway. For example, some receptor proteins respond to signals by activating intracellular protein kinases, and many signal transduction pathways ultimately regulate the synthesis, activity or stability of TFs. WRKY TFs play an important role in regulating the high-temperature stress responses of plants.

#### 4.2.1. ROS-Mediated Signaling Pathway

WRKY TFs have often been depicted as factors that increase ROS production in cells. High-temperature stress can cause the excessive accumulation of ROS in plants and produce oxidative stress. Recent studies have shown that some *WRKYs* are induced by ROS and participate in the ROS clearance transduction pathway [95,96]. Hydrogen peroxide treatment in *Arabidopsis* triggered higher expression of *AtWRKY30*, *AtWRKY75*, *AtWRKY48*, *AtWRKY39*, *AtWRKY6*, *AtWRKY53*, *AtWRKY22* and *AtWRKY8* [97], and *AtWRKY39* could respond to high-temperature stress. *OsWRKY42* has been portrayed as a negative regulator of oxidative stress, and overexpression of *OsWRKY42* in rice results in the high accumulation of ROS [98]. *TaWRKY10* in wheat overexpressing transgenic tobacco showed decreased accumulation of MDA, and low MDA was attributed to low rates of lipid peroxidation. Transgenic seedlings showed increased tolerance to oxidative stress due to the higher expression of *TaWRKY10* to resist damage induced by high temperature [99]. Babitha et al. [100] found that when subjected to oxidative stress, *Arabidopsis AtWRKY28* could increase resistance to ROS by regulating the expression of downstream related genes.

#### 4.2.2. Plant Hormone-Mediated Signaling Pathway

It is known that some WRKY TFs are involved in abscisic acid (ABA), jasmonic acid (JA), salicylic acid (SA), methyl jasmonate (MeJA) and other signal transduction pathways. Plant hormones regulate plant development and adaptation to environmental stress [101]. Through treatment with exogenous hormones, the corresponding signaling pathways can be activated, thereby improving the tolerance of plants to high temperature. However, it is still unclear how WRKY TFs coordinate and regulate these signaling pathways. First, WRKY TFs could activate plant hormone signals. In response to environmental threats, an *AtWRKY53* reporter gene in yeast could also activate ABA signaling and inhibit GA signaling; this activation and inhibition performance was achieved by means of transient expression in aleurone cells [22]. A study identified *Arabidopsis AtWRKY18*, *AtWRKY40* and *AtWRKY60* as ABA signal regulators that directly targeted the promoter regions of their respective coding genes, which was confirmed by chromatin immunoprecipitation and gel transfer experiments, and these three genes were found to be negative regulators of ABA signaling [102,103]. In addition, *Arabidopsis AtWRKY39* also positively regulates signaling pathways activated by SA and JA and regulates the response to high-temperature stress [104]. High-temperature treatment induces *AtWRKY39* transcription, whereas the SA and JA signaling pathways positively regulate *AtWRKY39* [60]. Similarly, *AtWRKY25*, *AtWRKY26* and *AtWRKY33*, feedback factors, promote the expression of *ETHYLENE-INSENSITIVE PROTEIN 2* (EIN2) and further activate the ethylene-mediated signal transduction pathway, thereby improving the high-temperature tolerance of plants [105]. Second, WRKY TFs respond to exogenous hormones to regulate plant high-temperature tolerance. *TaWRKY1* in wheat was slightly induced by high temperature and exogenous ABA, whereas *TaWRKY33* was resistant to high temperature and ABA, and MeJA responded quickly [52]. Dat et al. [106] reported that the external application of SA could effectively improve the tolerance of mustard seedlings to high temperature, although a high concentration did not show induction effects. This is the first report that external application could improve the resistance of the whole plant to high temperature. Silencing *TaWRKY70* expression led to greater sensitivity to stripe rust when wheat plants were treated with MeJA and high temperature (40 °C) stress [75].

#### 4.2.3. MAPK-Mediated Signaling Pathway

When plants are exposed to high-temperature stress, some receptors on their cell membranes are bound, which in turn activates the MAPK signaling pathway and transmits extracellular signals into the nucleus through phosphorylation or dephosphorylation reactions to inhibit or activate WRKY TF expression [107].

At present, many MAPKs that respond to high-temperature stress have been identified [108]. The first heat shock-activated MAPK identified in plants is HAMK in alfalfa [109]. MAPK, activated by high-temperature shock in tomato cells, could phosphorylate heat shock transcription factor a3 (HsFA3); alternating heat and cold conditions in rice could activate MAPK [110]; and temperature changes could affect the expression of *OsMRSMK2* in rice. When potatoes are exposed to high-temperature stress, *StMPK1* transcription is increased [111]. In *Arabidopsis*, flagellin flg22 could trigger the complete MEKK1-MKK4/5-MAPK3/6 MAPK cascade pathway and activate downstream *WRKY22/29* expression [112]. CaMAPK1 could interact with CaWKRY40 through bimolecular fluorescence interaction (BiFC) and participate in the response of pepper to high-temperature stress [113]. Studies have shown that wheat is sensitive to final high temperatures, leading to a sharp decline in grain quality and yield. Overall, in the defense mechanism network regulated by the MAPK cascade in plants, the MAPK cascade acts as the main switch (signal/sensory protein molecule) that regulates high-temperature stress by triggering WRKY TF and stress-related gene expression [93]. In summary, WRKY TFs may respond to high-temperature stress and thus form a network-like regulatory system, which reflects the complexity and diversity of the regulation of abiotic stress-related gene processes in plants (Figure 3).

## 5. Conclusions and Future Prospects

The normal growth and development of plants depend on the surrounding environmental conditions. WRKY TFs not only regulate the response of plants to high-temperature stress but also participate in specific signaling pathways. Therefore, it is of great significance to elucidate the complete mechanism by which WRKY TFs regulate plant responses to high-temperature stress. Based on the structure and function of WRKY, this review summarizes its role in regulating target genes and participating in signaling pathways. The mechanism by which plants tolerate high-temperature stress is very complicated. Plants undergo a series of metabolic pathways, and all metabolic pathways are regulated by genes. When plants are exposed to high-temperature stress, how do different MAPK signal cascades coordinate and divide the work? How do WRKY TFs regulate ROS production to adapt to stress? How do ROS levels in the high-temperature pathway change and induce WRKY TF expression? How do WRKY TFs participate in plant hormone regulation and signaling pathways? How do these proteins interact with each other in different signaling pathways? Currently, only a few WRKYs participate in plant responses to high-temperature stress, and significant research needs to be performed to fully elucidate the function of WRKYs. At present, research on the role of WRKY TFs in the regulation of plant responses to high-temperature stress is not sufficiently thorough, specifically at the transcriptional level. In the future, the function of WRKY proteins can be studied by synergizing transcription regulatory factors with other factors and through posttranscriptional and posttranslational modifications. With the help of biotechnology, such as high-throughput sequencing, genome-wide association studies and proteomics analysis, it can be used to understand different WRKY related networks and explain the response mechanism of plants to high-temperature stress. In view of this, the *WRKY* can be effectively used and applies to the molecular improvement of plants.

## Figures and Tables

**Figure 1 plants-10-02211-f001:**
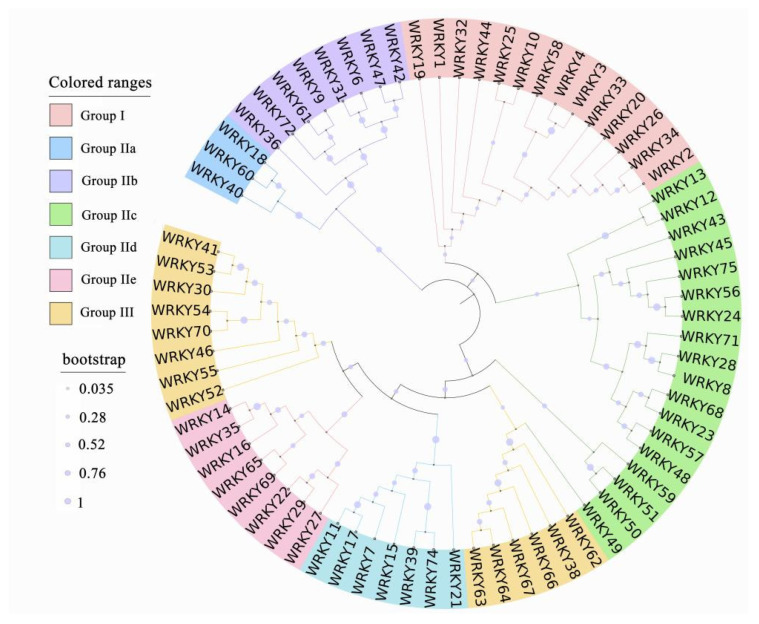
Phylogenetic tree of WRKY TFs from *Arabidopsis*. Through database searching, 72 ORFs encoding WRKY proteins were identified; MAGA7.0 software was used for multiple comparisons and the phylogenetic tree was generated with the iTOL program.

**Figure 2 plants-10-02211-f002:**
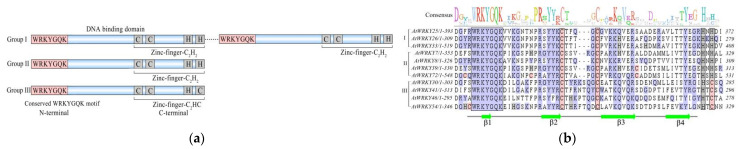
Structural features of WRKY TFs. (**a**) Structural features and classification of WRKY TFs. The pink areas indicate the N-terminal WRKYGQK sequence structure, the blue areas represent amino acid residues and the gray area indicates the head and end of the C-terminal zinc-finger structure. (**b**) Structure of WRKY TFs involved in the response to high-temperature stress in *Arabidopsis*. The black box indicates the conserved region and zinc-finger structure, and the green arrow indicates 4 β-sheets.

**Figure 3 plants-10-02211-f003:**
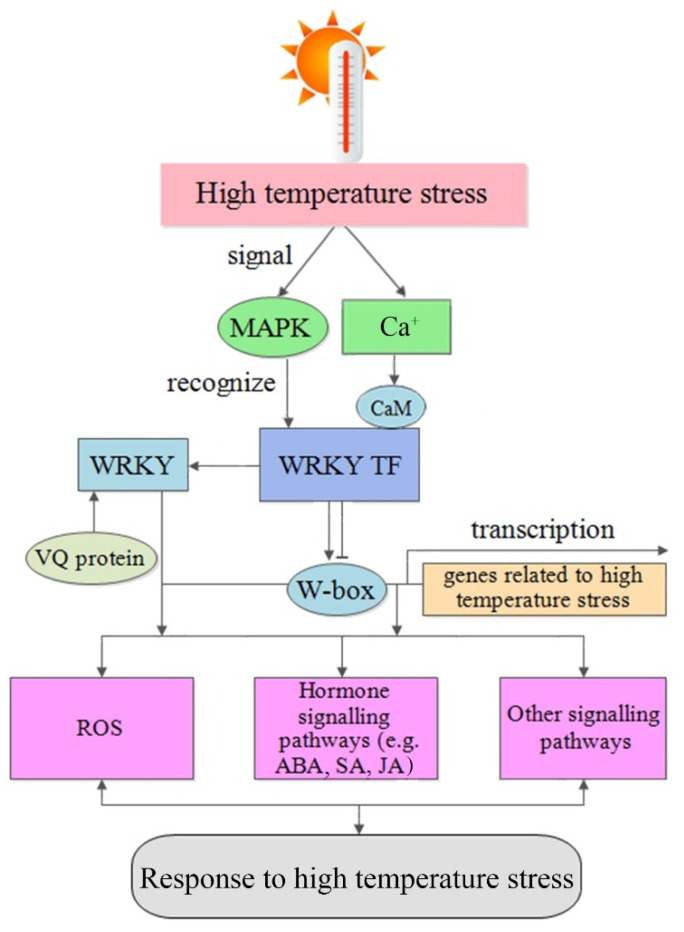
Model of WRKY TF responses to high-temperature stress in plants. The signaling pathway of plant WRKY TF responses to high-temperature stress is summarized. High-temperature environmental signals can trigger the activation of the MAPK signaling cascade and induce WRKY TFs to interact with target genes and related proteins. In addition, the signal can also promote an increase in intracellular Ca^2+^ levels, promote the binding of CaM to specific WRKY proteins, and change the transcriptional regulatory activity of WRKY proteins. Finally, the activation of the response to high-temperature stress occurs through ROS, hormones or other signaling pathways.

**Table 1 plants-10-02211-t001:** WRKY transcription factors partially respond to high-temperature stress.

No.	Gene	Plant Species	Research Method	References
1	*AtWRKY7*	*Arabidopsis thaliana*	Site-directed mutagenesis	[56]
2	*AtWRKY* *8*	*Arabidopsis thaliana*	Yeast two-hybrid	[57]
3	*AtWRKY25* *AtWRKY26* *AtWRKY33*	*Arabidopsis thaliana*	Constitutive expression, T-DNA insertion mutants	[58]
4	*AtWRKY30*	*Triticum aestivum*	Overexpression	[59]
5	*AtWRKY39*	*Arabidopsis thaliana*	Overexpression, knockout mutation	[60]
6	*AtWRKY* *41*	*Arabidopsis thaliana*	Gene silencing	[61]
7	*AtWRKY46*	*Arabidopsis thaliana*	Overexpression	[62]
8	*AtWRKY54*	*Arabidopsis thaliana*	Expression analysis, transcriptome and metabolome analysis	[63]
9	*AtWRKY72*	*Arabidopsis thaliana*	Yeast two-hybrid bimolecular fluorescence complementation, T-DNA insertion mutant	[64]
10	*CaWRKY27*	*Capsicum annuum*	Overexpression, gene silencing	[65]
11	*CaWRKY40*	*Capsicum annuum*	Overexpression, gene silencing	[66]
12	*CsWRKYs*	*Camellia sinensis*	Transcriptomics analysis	[67]
13	*CsWRKY46*	*Cucumis sativus*	Overexpression, VIGS	[68]
14	*GhWRKY21*	*Gossypium hirsutum*	VIGS, overexpression, yeast one-hybrid	[69]
15	*HaWRKY6*	*Helianthus annuus*	Expression analysis, northern blotting	[70]
16	*JrWRKY6* *JrWRKY53*	*Juglans regia*	Expression analysis	[71]
17	*NtWRKY6*	*Nicotiana tabacum*	Ectopic expression	[72]
18	*OsWRKY11*	*Oryza sativa*	Fusion with HSP101 promoter	[53]
19	*OsWRKY77*	*Oryza sativa*	Overexpression	[73]
20	*PtWRKY13* *PtWRKY50*	*Populus tomentosa*	Expression analysis, transcriptome and metabolome	[74]
21	*TaWRKY1* *TaWRKY33*	*Triticum aestivum*	Transcriptomics analysis	[52]
22	*TaWRKY30*	*Triticum aestivum*	Overexpression	[59]
23	*TaWRKY70*	*Triticum aestivum*	Gene silencing	[75]
24	*ZmWRKY106*	*Zea mays*	Expression analysis, transcriptome analysis	[76]

## Data Availability

The data presented in this study are available in article.

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
