# Peer review of "WRKY Transcription Factor Response to High-Temperature Stress"

_plants, 2021, doi:10.3390/plants10102211_

Round 1
Reviewer 1 Report
The manuscript entitled "WRKY transcription factor response to high temperature stress" is a good review where Authors compile different studies in order to highlight the relevance of WRKY TFs in relation to plant response to high temperature.
The manuscript presents a large amount of citations to include different aspects (WRKY interactions with other proteins, other WRKYs, other downstream genes…) in which WRKYs respond to high temperature stress. Despite the fact that there is a manuscript published in 2020 in Plants, mentioned by Authors, where the function and mechanism of WRKY TFs in abiotic stress responses of plant are studied; this manuscript is focused only in response to high temperatures and so, is a more- in-depth review in this aspect.
Manuscript is well written, it is easily readable and I would suggest publishing it.
-Line 108-110: Please correct “Giardia” and “Chlamydomonas” with upper case and Dictyostelium discoideum also in italics.
-I would suggest increasing the size/quality of the figures to make them easier to read
Author Response
Dear reviewer,
Thank you very much for your approval of the paper.
Point 1:-Line 108-110: Please correct “Giardia” and “Chlamydomonas” with upper case and Dictyostelium discoideum also in italics.
Response 1:We have corrected it, see line 119.
Point 2:-I would suggest increasing the size/quality of the figures to make them easier to read
Response 2:We sincerely thank you for raising this question, we have increased the size/quality of the figures in the figure, please see the attachment.
Point 3:English language and style are not fine/minor spell check required
Response 3:We have made some changes to the language and style of the paper, marked in revision mode, please see the attachment.

Reviewer 2 Report
The paper “WRKY transcription factor response to high temperature stress” by Zhuoya Cheng et al. is the review article as the title describes. The theme of the review is interesting and important and the authors collected a modest number of papers. However, because the structure of the article is not good, it is very hard to read and understand. Authors should reconsider the arrangement of the whole text.
- for example
P3: expression and function of WRKY are mixing. And the induction of expression by other stresses than high temperature is described randomly.
(The gene induction by high temperature and the addition of tolerance to high temperature by overexpressing gene is different).
Phylogenetic tree (Figure 2) should be first (structure and classification)
There are too many things to be revised and I cannot state all of them.
I could not understand what the authors want to state about the relation between WRKY and high temperature.
Please read the whole article and reconsider the arrangement and organization of the article.
Reviewer 3 Report
This is a short review focusing on WRKY transcription factors (TFs) and their role in response to heat stress. I felt this was a timely and useful review and generally well written. Most of my comments are “cosmetic”
Line 41: “regulating adversity”? The TFs are not really regulating adversity, but rather response to stresses…correct?
Line 51-53: the way this sentence reads seems a little awkward. Doesn’t the ROS lead to the reprogramming?
Line 72-74: This reads as though the WRKY domain is inducing a change in the DNA structure, that then allows the TF to bind DNA. Is this correct?
Line 108 – Please capitalize and italicize scientific names.
Line 120 references Figure 1. I do not see how this Figure is showing roles in high temp resistance. Also I wondered if it is possible to have some sort of “scale” on figure 1. When I first looked at it, it seems the zinc finger may be quite C-terminal to the WRKY (and part of this because the image I got in my head reading lines 62-63 was that the WRKY is N-terminal and the zinc finger is C-terminal – I now understand that they mean WRKY is at the N-terminal end of the ~60aa DNA binding domain, whereas the Zn-finger is at the C-term. End of this domain… correct? Overall, within a WRKY TF, is the WRKY domain N-terminal, C- or anywhere?
Line 127 – should the potato WRKY’s be StWRKY (rather than Sc)
Line 129 – what do they mean by wild sage allele?
Table 1 – So these are WRKY’s where there is evidence of response to high temp stress, but it seems to be a partial list as others are mentioned in the text but not in the table (for example TaWRKY30). So how complete is this table?
Line 171 reads “WRKY can regulate its own expression” – OK we are talking about a pretty big family. So do you mean “some WRKYs can regulate their own expression”?
I wondered in section 4.1 if any genome wide studies were done? It seems the answer is yes and they cite references, but I wondered if they wanted to comment.
Line 174 – consider an edit to WRKY interacts with other proteins.
Section 5 – are WRKY’s phosphorylated?
Line 249: WRKY08 or WRKY8 or WRKY80?
Line 278 – generally wild type genes are in Caps and italicized (at least for Arabidopsis). So, please consider “ETHYLENE-INSENSITIVE PROTEIN 2 (EIN2)”. Please check throughout (it is easy to miss!).
Reference 49 – I think it should be Zhang, C (rather than Chao, Z).
I hope these suggestions are helpful.
Author Response
Dear reviewer:
Thank you for your contribution, and we sincerely thank you for raising these questions.
Point 1:English language and style are not fine/minor spell check required
Response 1: We have made some changes to the language and style of the article, marked in revision mode, please see the attachment.
Point 2: Line 41: “regulating adversity”? The TFs are not really regulating adversity, but rather response to stresses…correct?
Response 2:We agree with you and made corrections, see line 43.
Point 3:Line 51-53: the way this sentence reads seems a little awkward. Doesn’t the ROS lead to the reprogramming?
Response 3: The high temperature stress induction pathway includes signal transduction components and also induces genome-wide transcriptional reprogramming, thereby inducing other protective mechanisms.
Point 4:Line 72-74: This reads as though the WRKY domain is inducing a change in the DNA structure, that then allows the TF to bind DNA. Is this correct?
Response 4: The WRKY domain forms a DNA groove by itself for binding to DNA. The DNA binding function is mainly induced by the C-terminal zinc finger domain.
Point 5:Line 108 – Please capitalize and italicize scientific names.
Response 5: This problem has been corrected, see line 119.
Point 6:Line 120 references Figure 1. I do not see how this Figure is showing roles in high temp resistance. Also I wondered if it is possible to have some sort of “scale” on figure 1. When I first looked at it, it seems the zinc finger may be quite C-terminal to the WRKY (and part of this because the image I got in my head reading lines 62-63 was that the WRKY is N-terminal and the zinc finger is C-terminal – I now understand that they mean WRKY is at the N-terminal end of the ~60aa DNA binding domain, whereas the Zn-finger is at the C-term. End of this domain… correct? Overall, within a WRKY TF, is the WRKY domain N-terminal, C- or anywhere?
Response 6: Your question is very useful to us. We may not express it clearly in my pictures and articles. Figure 1 We want to illustrate the structure of some WRKY genes in Arabidopsis that respond to high temperature as an example. We have re-adjusted the diagram so that it can be read better (See Figure 2). In general, the DNA binding domain of WRKY is named WRKY domain, which is a polypeptide sequence composed of approximately 60 conserved amino acid residues, of which the N-terminal WRKYGQK is a highly conserved 7 amino acid residues among all members,C-terminal is a zinc finger structure.
Point 7:Line 127 – should the potato WRKY’s be StWRKY (rather than Sc)
Response 7: This problem has been corrected.
Point 8:Line 129 – what do they mean by wild sage allele?
Response 8: Did not pay attention in the previous writing, this part does not involve high temperature, so it is deleted.
Point 9:Table 1 – So these are WRKY’s where there is evidence of response to high temp stress, but it seems to be a partial list as others are mentioned in the text but not in the table (for example TaWRKY30). So how complete is this table?
Response 9: Thank you for your suggestion. We reviewed the article and adjusted the table. (See Table 1)
Point 10:Line 171 reads “WRKY can regulate its own expression” – OK we are talking about a pretty big family. So do you mean “some WRKYs can regulate their own expression”?
Response 10: WRKY TFs can be combined with their own promoters or other WRKY TFs promoters to realize self-regulation or cross-regulation network. We re-describe in the text, see line 159.
Point 11:I wondered in section 4.1 if any genome wide studies were done? It seems the answer is yes and they cite references, but I wondered if they wanted to comment.
Response 11: Some WRKY performed genome-wide predictions of target genes, and some high temperature-related examples are described in the article
Point 12:consider an edit to WRKY interacts with other proteins.
Response 12: Thank you very much for your comments. I think that the binding of WRKY to itself and other WRKY promoters belongs to the binding of downstream target genes. It is not the same as WRKY and WRKY protein binding.
Point 13:are WRKY’s phosphorylated?
Response 13: WRKY can be phosphorylated by MAPK and interacts with specific cis-elements in its promoter region to directly regulate the expression of a series of downstream genes
Point 14:Line 249: WRKY08 or WRKY8 or WRKY80?
Response 14: It is WRKY8 has been corrected
Point 15:Line 278 – generally wild type genes are in Caps and italicized (at least for Arabidopsis). So, please consider “ETHYLENE-INSENSITIVE PROTEIN 2 (EIN2)”. Please check throughout (it is easy to miss!).
Response 15: The full text has been checked, related issues have been corrected and marked with revision mode.
Point 16:Reference 49 – I think it should be Zhang, C (rather than Chao, Z).
Response 16: I agree with your opinion and correct it.

Reviewer 4 Report
The authors performed an interesting article analysis. The summarized material is valuable and interesting to the reader.
Author Response
Dear reviewer,
Thank you very much for taking the time to read this paper, and thank you for your support of this paper, I will try my best to improve it.
Round 2
Reviewer 2 Report
The revised paper “WRKY transcription factor response to high temperature stress” by Zhuoya Cheng et al. was improved and it became to be easier to read. However, there are still several parts to improved.
- The last part of introduction is confusing (L43-64)
- The authors mention that many transcription factors play important roles in response to adverse conditions and refer to (9). However, the reference (9) is another review of WRKY genes. Reference should be corrected.
- I cannot understand the sentence L58-61.
- I don’t understand the meaning of the insertion of the sentence L50-55 (example of WRKY function in the stress response?).
- The flow of the story should be like the following,
- many TFs are involved in the response to the abiotic stresses including high-temperature.
- WRKY family is one of the TF families involved in the response to the abiotic stresses.
(for example, maybe sentence like L50-51?)
- WRKY family is known to be involved in the high-temperature response.
- This review describes xxxx.
- Reference (9) is also the review of WRKY and it mentions the WRKY function in the high-temperature response. I think the author should stress the point of the review which is different from the previous review reference (9).
- I cannot understand the sentence L361-363.
